# Morphological, molecular and MALDI-TOF MS identification of ticks and tick-associated pathogens in Vietnam

**Ly Na Huynh**[1,2,3], **Adama Zan Diarra**[1,2], **Quang Luan Pham**[3], **Nhiem Le-Viet**[4], **Jean-Michel Berenger**[1,2], **Van Hoang Ho**[3], **Xuan Quang Nguyen**[3], **Philippe Parola**[1,2] *

**1** Aix Marseille Univ, IRD, AP-HM, SSA, VITROME, Marseille, France, **2** IHU-Méditerranée Infection, Marseille, France, **3** Institute of Malariology, Parasitology and Entomology, Quy Nhon (IMPE-QN), Vietnam, **4** School of Medicine and Pharmacy, The University of Da Nang (UD), Da Nang, Vietnam

* philippe.parola@univ-amu.fr

**Data Availability Statement:** All relevant data are within the manuscript and its Supporting Information files.

## Abstract

Matrix-assisted laser desorption/ionization time-of-flight mass spectrometry (MALDI-TOF MS) has been reported as a promising and reliable tool for arthropod identification, including the identification of alcohol-preserved ticks based on extracted leg protein spectra. In this study, the legs of 361 ticks collected in Vietnam, including 251 *Rhiphicephalus sanguineus* s.l, 99 *Rhipicephalus* (*Boophilus*) *microplus*, two *Amblyomma varanensis*, seven *Dermacentor auratus*, one *Dermacentor compactus* and one *Amblyomma* sp. were submitted for MALDI-TOF MS analyses. Spectral analysis showed intra-species reproducibility and inter-species specificity and the spectra of 329 (91%) specimens were of excellent quality. The blind test of 310 spectra remaining after updating the database with 19 spectra revealed that all were correctly identified with log score values (LSV) ranging from 1.7 to 2.396 with a mean of 1.982 ± 0.142 and a median of 1.971. The DNA of several microorganisms including *Anaplasma platys*, *Anaplasma phagocytophilum*, *Anaplasma marginale*, *Ehrlichia rustica*, *Babesia vogeli*, *Theileria sinensis*, and *Theileria orientalis* were detected in 25 ticks. Co-infection by *A. phagocytophilum* and *T. sinensis* was found in one *Rh.* (*B*) *microplus*.

## Author summary

Ticks are one of the important vectors and reservoirs of multiple pathogens infecting humans and animals such as bacteria, protozoans, viruses, and helminths. Nevertheless, studies on ticks and tick-borne infections remain limited in Vietnam. That said, serological and molecular evidence of tick infections in animals and humans have been reported on several occasions in Vietnam and Southeast Asia in recent decades. The identification of ticks and tick-associated diseases has an important role to play in epidemiological investigation and in assessing the risks of disease transmission to humans and animals. Recently, MALDI-TOF MS has been used as an innovative tool for the rapid and accurate identification of alcohol-preserved ticks based on proteins from extracted legs. This procedure represents a time-cost saving and does not require expert knowledge. This goal of

**Funding:** This study was supported by the Institut Hospitalo-Universitaire (IHU) Méditerranée Infection, the National Research Agency under the "Investissements d'avenir" programme, reference ANR-10-IAHU-03, the Région Provence Alpes Côte d'Azur and European ERDF PRIMI funding. LNH received a grant of PhD scholarship from IHU Méditerranée Infection. The funders had no role in study design, data collection and analysis, decision to publish, or preparation of the manuscript.

**Competing interests:** The authors have declared that no competing interests exist.

this study was to assess the efficiency and reliability of MALDI-TOF MS for the identification of alcohol-preserved ticks collected in Vietnam and to determine the presence of their relative pathogens. Our study revealed 97% correspondence between morphological and MALDI-TOF MS identification. The detected microorganisms that were confirmed by sequencing belonged to the Anaplasmataceae and Piroplasmida families. These findings suggested that ticks and tick-associated pathogens are likely to pose challenges to public and veterinary health in Vietnam.

## Introduction

Ticks have been incriminated as the second most important vectors of human and animal infectious pathogens in the world after mosquitoes [1] and are able to transmit a wide range of pathogens, including bacteria, protozoans, viruses, and helminths [2]. In Southeast Asia (SEA), there are 104 known tick species, representing 12 genera, which is approximately 12% of all recognised and classified species [3]. Among them, *Rhipicephalus sanguineus* sensu lato (s.l.) are the most common ticks that parasitise dogs in SEA. These ticks are the ectoparasite vectors of bacterial and protozoal pathogens that can be transmitted to animals [4] and humans [5]. *Rhipicephalus* (*Boophilus*) *microplus* is an important vector of livestock pathogens [6]. *Amblyomma* (formerly *Aponomma*) *varanensis*, *Dermacentor auratus*, and *Dermacentor compactus* may act as vectors of infectious agents (e.g. *Rickettsia* spp., *Anaplasma* spp., *Ehrlichia* spp., *Borrelia* spp., *Babesia* spp. and *Theileria* spp.) to humans, and to domestic and wild animals in Malaysia, Laos, Thailand, and Vietnam [7–10].

In Vietnam, the agricultural sector makes up one-third of the developing nation's economy [11], and livestock represents the second biggest contribution to household incomes after crop growing [12]. Despite the perceived food and economic benefits of livestock production, the country is potentially faced with challenges such as the emergence and re-emergence of zoonotic diseases, which can cause huge losses [13, 14]. One such example is the risk of infectious diseases spreading through the large number of dogs that are illegally imported into Vietnam from neighbouring countries for food consumption without any veterinary controls [15, 16]. In 2014, an outbreak of oriental theileriosis, which causes abortion and death, in imported cattle from Australia to Vietnam was associated with *Theileria orientalis* [17]. The serological detection of both *Babesia bovis* and *Babesia bigemina* parasite species transmitted by ticks has also been reported in cattle imported from Thailand [18].

Limited data is available on ticks and tick-associated pathogens in Vietnam. Nevertheless, 48 species of nine tick genera have been reported by Kolonin [19] and recently two new species of ticks of the genus *Dermacentor* (*Dermacentor limbooliati* and *Dermacentor filippovae*) have been described by Apanaskevich [9, 20]. Also in Vietnam, some tick-borne microorganisms have been reported in ticks and animals [19, 21–23], more precisely in *Hepatozoon canis*, *Ehrlichia canis*, and *Babesia vogeli* ticks [24].

In recent years, several studies have focused on acarology in Vietnam [4, 10, 25]. The correct identification of ticks is a crucial step in distinguishing tick vectors from non-vectors. The lack of reference data and standard taxonomic keys specific to Vietnamese tick species makes the morphological identification of Vietnamese ticks difficult or almost impossible. The morphological identification of tick species therefore remains a challenge for Vietnamese researchers [19]. Molecular tools have been used to overcome the limitations of morphological identification [26]. However, there are several drawbacks to these tools, which are time-consuming, expensive, and require primer-specific targeting [27–29].

Recently, the MALDI-TOF MS method has been proposed as an alternative and innovative tool to overcome the limitations of the above two methods in arthropod identification [30]. Since then, studies in several laboratories have demonstrated that MALDI-TOF MS is a remarkably robust tool for identifying many species of arthropod vectors and non-vectors [30]. The aim of this study was to identify tick species collected from domestic and wild animals in Vietnam and their associated pathogens using morphological, MALDI-TOF MS and molecular tools.

## Materials and methods

### Ethics statement

Ethical approval was obtained from the Institute of Malariology, Parasitology, and Entomology, Quy Nhon (IMPE-QN) on behalf of the Vietnamese Ministry of Health (approval no: 401/VSR-CT-2010, 333/CT-VSR-2018). Permission was obtained from the communal authorities for wild animals that were not listed in the Red Data Book of Vietnam, and agreement was obtained from the owners of cows, goats, and dogs.

### Tick collection and morphological identification

Ticks were collected in four provinces: Quynh Luu (19°13' N; 105°60' E) District, Nghe An Province; Nam Giang (15°65' N; 107°50' E) District, Quang Nam Province; Van Canh (13°37' N; 108°59' E) District, Binh Dinh Province; and Khanh Vinh (12°16' N; 108°53' E) District, Khanh Hoa Province in Vietnam in September 2010, between April and September 2018. The map of Vietnam showing the collection sites was made with QGIS version 3.10 and the Vietnamese layers were downloaded from DIVA-GIS at the following link: https://www.diva-gis.org/datadown (Fig 1A). All engorged and non-engorged ticks were collected from the skin of domestic animals (cows, goats, and dogs) and wild animals (pangolins, wild pigs) using forceps. Ticks from wild animals were collected in a collaborative manner by rangers and trained care personnel from the Wildlife Rescue, Conservation and Development Center. Ticks were morphologically identified first at species level using dichotomous keys [9, 31] by an entomological team from the Institute of Malariology, Parasitology and Entomology, Quy Nhon,

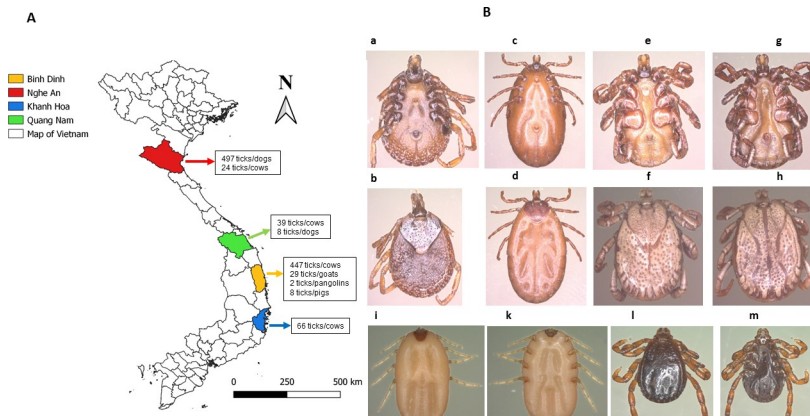

**Fig 1.** Map of Vietnam showing tick collection sites realised with QGIS version 3.10, the layers have been uploaded to the DIVA-GIS website: **https://www.diva-gis.org/datadown** (A); Morphologically, the 70% alcohol tick-preserved species were collected in Vietnam over a period of 10 years: *Amblyomma varanensis* [♀: **a, b**]; *Amblyomma* sp. [♀: **c, d**]; *Dermacentor auratus* [♂: **e, f**]; *Dermacentor compactus* [♂: **g, h**]; and approximately 2 years: *Rhipicephalus* (*B*) *microplus* [♀: **I, k**]*; Rhipicephalus sanguineus* s.l [♂: **l, m**] **(B)**.

Vietnam (IMPE-QN). Ticks from the same host were counted and placed in the same tube containing 70% v/v alcohol, before being sent to the Institut Hospitalo-Universitaire (IHU) Méditerranée Infection in Marseille, France for MALDI-TOF MS and molecular analysis. In Marseille, the morphological identification of ticks was verified by two specialists in morphological identification of ticks using a magnifying glass (Zeiss Axio Zoom.V16, Zeiss, Marly le Roi, France) and dichotomous keys. Morphological identification was carried out only if all discriminating characters had been observed.

## Tick dissection and sample preparation

Ticks were individually removed from the alcohol and were rinsed and dissected with a sterile surgical blade, as previously described [32]. The four legs of each tick and the half part without legs were submitted for MALDI-TOF MS and molecular biology analysis, respectively. The remaining parts with legs were frozen and stored as samples for any further research.

## DNA extraction and molecular identification of ticks

DNA from each half-tick or legs (for ticks from which we did not obtain sequences with half-tick DNA) was individually extracted using an EZ1 DNA tissue kit (Qiagen), according to the manufacturer's recommendations, as previously described [33]. DNA was monitored with Nanodrop 1000 Spectrophotometer (Thermo Fisher Scientific, Wilmington, USA) and either immediately used or stored at -20˚C until use.

DNA from ticks was submitted to standard PCR in an automated DNA thermal cycle to amplify a 465-base pair (bp) fragment of the mitochondrial *16S* DNA gene, as described previously [34]. The *12S* tick gene, amplifying about 405-bp of the mitochondrial DNA fragment, was used for all specimens for which we did not have a sequence with the *16S* gene. DNA from *Rh. sanguineus* s.l., reared in our laboratory, was used as a positive control. Purified PCR products were sequenced as previously described [34]. The obtained sequences were assembled and analysed using the ChromasPro software (version 1.7.7) (Technelysium Pty. Ltd., Tewantin, Australia), and were then blasted against the reference sequences available in GenBank (http://blast.ncbi.nlm.nih.gov/).

## MALDI-TOF MS analysis

**Sample preparation.** The four legs of each tick were first put into an Eppendorf tube and dried overnight at 37˚C and then put into an Eppendorf tube with 40 μL of high-performance liquid chromatography (HPLC) grade water and incubated overnight at 37˚C. The legs were then crushed in a mix of 20 μL of 70% (v/v) formic acid (Sigma) and 20 μL of 50% (v/v) acetonitrile (Fluka, Buchs, Switzerland), with glass beads (Sigma, Lyon, France), as described previously [35]. The crushed legs were centrifuged and 1 μL of the supernatant of each sample was deposited in quadruplicate onto a MALDI-TOF MS steel plate (Bruker Daltonics, Wissembourg, France). After drying at room temperature, 1μL of matrix solution composed of a saturated solution of α-cyano-4-hydroxycynnamic acid (Sigma, Lyon, France), 50% acetonitrile (v/v), 2. 5% trifluoroacetic acid (v/v) (Aldrich, Dorset, United Kingdom), and high performance liquid chromatography (HPLC) grade water was added [36]. The target plate was air-dried one more at room temperature before being introduced into the Microflex LT MALDI-TOF Mass Spectrometer (Bruker Daltonics, Germany) for analysis. The quality of the matrix, sample loading, and performance of the MALDI-TOF MS device were controlled using the legs of a *Rh. sanguineus* s.l. reared in our laboratory as a positive control.

**MALDI-TOF MS parameters, spectral analysis and reference database creation.** The spectral profiles obtained from the tick legs were visualised using a Microflex LT MALDI-TOF

mass spectrometer with FlexControl software (version 3.3, Bruker Daltonics). The setting parameters of the MALDI-TOF MS apparatus were identical to those previously used [32].

The FlexAnalysis v.3.3 software was used to evaluate spectral quality (smoothing, baseline subtraction, peak intensities). MS spectra reproducibility was assessed by comparing the average spectral profiles (MSP, main spectrum profile) obtained from the four spots of each tick leg, according to species, using MALDI-Biotyper v3.0 software (Bruker Daltonics) [37]. MS spectra reproducibility and specificity were assessed based on a principal component analysis (PCA) and cluster analysis (MSP dendrogram). PCA was performed using ClinProTools v2.2 with the manufacturer's default settings. Cluster analysis was performed based on a comparison of the MSP given by MALDI-Biotyper v3.0. software with clustering according to protein mass profile (i.e., their mass signals and intensities) [37].

Based on the morphological identification, eight and seven reference spectra of *Rh. sanguineus* and *Rh. (B) microplus*, respectively, were added to our MALDI-TOF MS database. However, two, one, and one spectra of *D. auratus*, *Am. varanensis*, *D. compactus*, respectively, which were only identified morphologically by three tick identification specialists, were also added to our MALDI-TOF MS database. To create a database, reference spectra (MSP, Main Spectrum Profile) were created by combining the results of spectra from specimens of each species using the automated function of the MALDI-Biotyper v3.0 software (Bruker Daltonics). MSPs were created based on an unbiased algorithm using peak position, intensity, and frequency data [38]. Four tick species that could not be identified by molecular biology were temporarily added into the MS reference database to identify the remaining specimens from the same species.

**Blind test for tick identification.**   A blind test was performed with the remaining tick specimens not included in our MALDI-TOF MS database after the database had been upgraded with 19 MS spectra from specimens of the five tick species to determine their identification. The reliability of tick species identification was estimated using the log score values (LSVs) obtained from the MALDI-Biotyper software, which ranged from 0 to 3. These LSVs correspond to the degree of similarity between the MS reference spectra in the database and those submitted to blind tests. An LSV was obtained for each spectrum of the samples tested. According to one previous study [37], an LSV of at least 1.8 should be obtained to be considered reliable for species identification.

**Detection of microorganisms.**   Quantitative PCR (qPCR) was performed for screening microorganisms using specific primers and probes targeting Anaplasmataceae, Piroplasmida, *Borrelia* spp., *Bartonella* spp., *Coxiella burnetii*, and *Rickettsia* spp. PCR reactions were performed according to the manufacturer's instructions, using a CFX96 Touch detection system (Bio-Rad). qPCR amplification was performed using the thermal profile described previously [39]. The DNA of *Rickettsia montanensis*, *Bartonella elizabethae*, *Anaplasma phagocytophilum*, *Coxiella burnetii*, *Borrelia crocidurae*, and *Babesia vogeli* were used as a positive control and DNA from *Rh. sanguineus* s.l from our laboratory, which were free of bacteria, were used as negative controls. The samples were considered to be positive when the cycle threshold (Ct) was strictly less than 36 [40].

All samples that were positive following qPCR were submitted to standard PCR and sequencing to identify the microorganism species. For the *Rickettsia* sp. positive sample, we first used a primer targeting a 630-bp fragment of the *OmpA* gene [35] and then another targeting a 401-bp fragment of the *gltA* gene [33]. Samples which were Anaplasmataceae positive following qPCR were subjected to amplifying and sequencing of a 520-bp fragment of the *23S* rRNA gene [33]. Samples which were Piroplasmidae positive following qPCR were subjected to amplifying and sequencing of a 969-bp fragment of the *18S* rRNA [41]. Samples which were *Borrelia* sp. positive following qPCR was subjected to amplifying and sequencing of a 344-bp

**Table 1. Target amplified and used for qPCR and standard PCR.**

| Microorganisms | Targeted sequence | Primers (5'-3') and Probes (Used for qPCR Screening or Sequencing) | References |
|---|---|---|---|
| **Anaplasmataceae** | 23S | f_TGACAGCGTACCTTTTGCAT<br>r_GTAACAGGTTCGGTCCTCCA<br>p_6FAM-GGATTAGACCCGAAACCAAG | [108] |
| | 23S (520-bp) | f_ATAAGCTGCGGGGAATTGTC<br>r_TGCAAAAGGTACGCTGTCAC | |
| **Piroplasmida** | 5.8S | f_AYYKTYAGCGRTGGATGTC<br>r_TCGCAGRAGTCTKCAAGTC<br>p_FAM-TTYGCTGCGTCCTTCATCGTTGT-MGB | [39] |
| | 18S (969-bp) | f1_GCGAATGGCTCATTAIAACA<br>f4_CACATCTAAGGAAGGCAGCA<br>f3_GTAGGGTATTGGCCTACCG*<br>r4_AGGACTACGACGGTATCTGA* | |
| **Rickettsia spp.** | gltA(RKND03) | f_GTGAATGAAAGATTACACTATTTAT<br>r_GTATCTTAGCAATCATTCTAATAGC<br>p_6FAM-CTATTATGCTTGCGGCTGTCGGTTC | [109] |
| | gltA (401-bp) | f_ATGACCAATGAAAATAATAAT<br>r_CTTATACTCTCTATGTACA | [110] |
| | OmpA (630-bp) | 70_ATGGCGAATATTTCTCCAAAA<br>701_GTTCCGTTAATGGCAGCATCT<br>180_GCAGCGATAATGCTGAGTA* | [1] |
| **Borrelia spp.** | ITS4 | f_GGCTTCGGGTCTACCACATCTA<br>r_CCGGGAGGGGAGTGAAATAG<br>p_TGCAAAAGGCACGCCATCACC | [111] |
| | flaB (344-bp) | f_TGGTATGGGAGTTTCTGG<br>r_TAAGCTGACTAATACTAATTACCC | |
| **Bartonella spp.** | ITS2 | f_GATGCCGGGGAAGGTTTTC<br>r_GCCTGGGAGGACTTGAACCT<br>p_GCGCGCGCTTGATAAGCGTG | [112] |
| **Coxiellia burnetii** | IS30A | f_CGCTGACCTACAGAAATATGTCC<br>r_GGGGTAAGTAAATAATACCTTCTGG<br>p_CATGAAGCGATTTATCAATACGTGTATG | [113] |

Abbreviation

*, used for sequencing only.

fragment of the *flaB* gene [42]. The primers and probes used in this study are listed in Table 1. The obtained sequences were assembled and analysed using the ChromasPro software (version 1.7.7) (Technelysium Pty. Ltd., Tewantin, Australia), and were then blasted against the reference sequences available in GenBank (http://blast.ncbi.nlm.nih.gov/). The method used for phylogenetic tree analysis was the neighbour-joining (NJ) method with 1,000 replicates. DNA sequences were aligned using MEGA software version 7.0 (https://www.megasoftware.net/). The various statistical analyses were performed using R software version 3.4 (R Development Core Team, R Foundation for Statistical Computing, Vienna, Austria) and ggplot packages were used to perform the graphics.

## Results

### Tick collection and morphological identification

A total of 1120 ticks including 334 (30%) engorged ticks were collected in four provinces of Vietnam: Nghe An, Quang Nam, Binh Dinh, and Khanh Hoa. Morphologically, ticks were identified as belonging to six species (Fig 1A), including 935 (83.5%) *Rh. sanguineus* s.l. collected from dogs, 174 (15.5%) *Rh.* (*B*) *microplus*) from cows and goats, seven (0.6%) *D. auratus*

**Table 2. The number of tick species used for MALDI-TOF MS analysis, creation of the MS reference spectra creation, and molecular biology confirmation.**

| Morphological identification | Number submitted for molecular ID* | Molecular ID* (%identity; GenBank accession number) | Number of good spectra/ tested | Number of spectra added to DB$ | MADI-TOF MS ID* (number identified) | LSVs & [Low-High] |
|---|---|---|---|---|---|---|
| *Rhipicephalus sanguineus* s.l | 8 | *Rh. sanguineus* s.l (99.75–100%; MG651947, MG793434, KX632154) | 241/251 | 8 | *Rh. sanguineus* s.l (233) | [1.7–2.351] |
| *Rhipicephalus (B) microplus* | 7 | *Rh. (B) microplus* (100%; MN880401, MT462222, EU918187) | 78/99 | 7 | *Rh. (B) microplus* (71) | [1.705–2.346] |
| *Amblyomma varanensis* | 1 | not identified | 1/2 | 1 | NA | NA |
| *Amblyomma* sp. | 1 | not identified | 1/1 | 0 | *Am.varanensis* (1) | 1.857 |
| *Dermacentor auratus* | 7 | not identified | 7/7 | 2 | *D. auratus* (5) | [1.949–2.396] |
| *Dermacentor compactus* | 1 | not identified | 1/1 | 1 | NA | NA |
| **Total** | **25** | | **329/361** | **19** | **310** | |

*Identification

& Range of log score values

$ Database.

from pangolins, two (0.2%) *Am. varanensis* from wild pigs, and one (0.1%) *D. compactus* and one (0.1%) *Amblyomma* sp. from a pangolin (Table 2). *Rhipicephalus sanguineus* s.l. and *Rh. (B) microplus* were collected between April and September 2018. The other ticks were collected in September 2010. The different specimens that could not be identified by molecular biology are shown in the pictures in Fig 1B that we took using a magnifying glass (Zeiss Axio Zoom. V16, Zeiss, Marly le Roi, France).

## Molecular identification of ticks

To confirm our morphological identification, 25 tick specimens were submitted to molecular analysis using the *16S* rDNA gene, including eight specimens of *Rh. sanguineus* s.l., seven *Rh. (B) microplus*, seven *D. auratus*, one *Am. varanensis*, one *D. compactus* and one *Amblyomma*

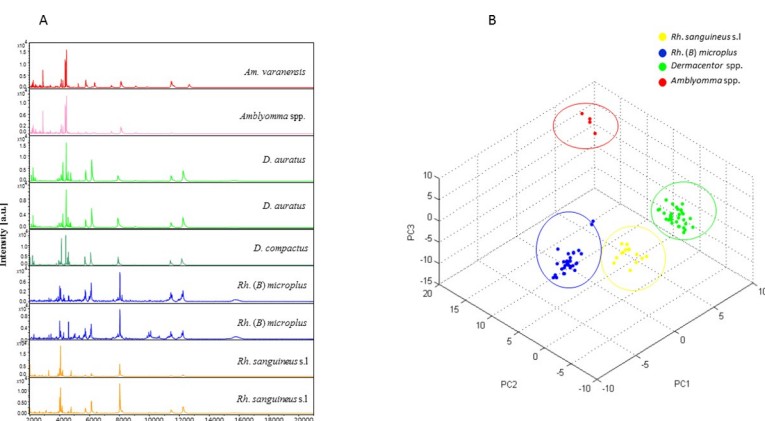

**Fig 2. Comparison of MALDI-TOF MS spectra from the legs of six tick species collected in Vietnam.** The MS spectra revealed intra-species reproducibility and inter-species specificity (A); The MS spectra were compared by Principal Component Analysis (B); a.u., arbitrary units; m/z, mass-to-charge ratio.

sp. Sequences were obtained only for the specimens of *Rh. sanguineus* s.l. and *Rh.* (*B*) *microplus*. BLAST analysis indicated that obtained sequences from *Rh. sanguineus* s.l. were 99.75 to 100% identical to the corresponding sequences of *Rh. sanguineus* s.l. (Genbank: MG651947, MG793434, KX632154) and those obtained from *Rh.* (*B*) *microplus* were 100% identical to the corresponding sequences of *Rh.* (*B*) *microplus* (Genbank: MN880401, MT462222, EU918187). Unfortunately, for the specimens morphologically identified as *D. auratus*, *Am. varanensis*, *Amblyomma* sp. and *D. compactus*, we were unable to amplify any DNA from the half-tick or legs of these tick species with PCR targeting part of the two genes (*16S* and *12S* rDNA), despite the fact that the nanodrop had indicated that the amount of DNA contained in these samples was 7.8 to 19.4 ng/μl.

## MS reference spectra analysis

The legs of 361 specimens, including 251 morphologically identified as *Rh. sanguineus* s.l., 99 *Rh.* (*B*) *microplus*, seven *D. auratus*, two *Am. varanensis*, one *Amblyomma* sp. and one *D. compactus* were randomly selected and subjected to MALDI-TOF MS analysis. Visualisation of MS spectra from all specimens using FlexAnalysis v.3.3 software showed that 91% (329) of specimens had excellent quality spectra (peak intensity > 3,000 a.u., no background noise and baseline subtraction correct) (Figs 2A and S1 and Table 2). The MS spectra of different specimens showed intra-species reproducibility and inter-species specificity, as confirmed by PCA (Figs 2B and 3B) and dendrogram (Fig 3A) analysis. PCA and dendrogram analysis showed that all specimens of the same species were grouped together or were on the same branches. Additionally, at the genus level, all specimens from the same genus were also gathered in the same part of dendrogram (Fig 3A).

## MALDI-TOF MS tick identification by blind test

The 310 MS remaining spectra of excellent quality, including 233 *Rh. sanguineus* s.l, 71 *Rh.* (*B*) *microplus*, five *D. auratus* and one *Amblyomma* sp. were queried against our reference spectra database upgraded with eight *Rh. sanguineus* s.l. and seven *Rh.* (*B*) *microplus* which were morphologically and molecularly identified, and two *D. auratus*, one *Am. varanensis* and one *D. compactus* identified only morphologically. The spectra of the ticks introduced in the

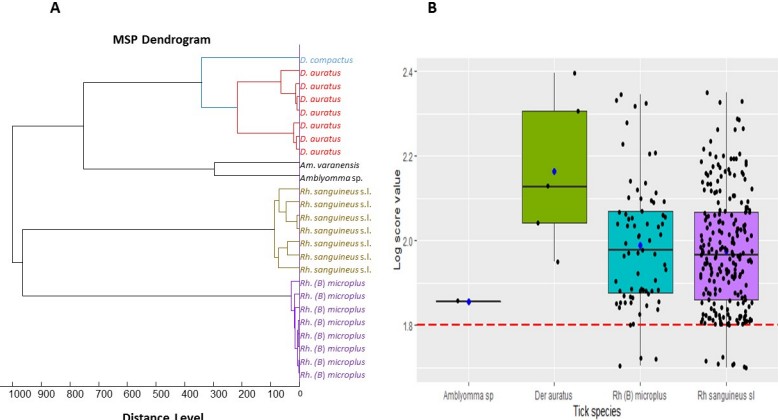

**Fig 3. Comparison of MALDI-TOF MS spectra from the legs of six alcohol-preserved tick species collected in Vietnam and stored for different periods of time.** The dendrogram was built using between one and eight representative MS spectra from six distinct tick species (A). The MS spectra of different specimens showed intra-species reproducibility and inter-species specificity as confirmed by PCA (B).

MALDI-TOF MS database have been deposited on the website of the University Hospital Institute (UHI) under the following DOI: https://doi.org/10.35088/rbqp-g648. The blind test revealed that 100% (233) of *Rh. sanguineus* s.l. specimens were correctly identified as *Rh. sanguineus* s.l. with LSVs ranging from 1.7–2.351 with a mean of 1.976 ± 0.137, 100% (71) of *Rh.* (*B*) *microplus* identified with LSVs ranging from 1.705–2.346 with a mean of 1.989 ± 0.148 and 100% (five) *D. auratus* with LSVs of 1.949–2.396 with a mean of 2.164 ± 0.149 (Table 2). The tick identified morphologically as *Amblyomma* sp. was identified by MALDI-TOF MS as *Am. varanensis* (LSV = 1.857) (Table 2). All our specimens were identified with LSVs ranging from 1.7–2.396 with a mean of 1.982 ± 0.142 and a median of 1.971, and 97% (301) had LSVs >1.8, which is considered the threshold for identification (Fig 3B). No blind test was performed for *D. compactus* because of the low number of specimens.

## Detection of microorganisms in ticks

A total of 361 ticks, including 260 (72%) non-engorged and 101 (28%) engorged ticks, were examined for the DNA of six microorganisms using qPCR. Thirty-nine (10.8%) were positive for at least one of the microorganisms, including Anaplasmataceae, *Rickettsia* spp, *Borrelia* spp. and Piroplasmida (Table 3). Notably, two *Rh.* (*B*) *microplus* specimens were co-infected with both Anaplasmataceae and Piroplasmida. No samples were positive for *C. burnetii* or *Bartonella* spp.

DNA from bacteria of the Anaplasmataceae family were detected in 18/361 (5%) of ticks by qPCR. The DNA of bacteria belonging to the Anaplasmataceae family was found in 13 (72%) *Rh.* (*B*) *microplus* and five (28%) *Rh. sanguineus* s.l. We successfully obtained seven (40%) sequences all from *Rh.* (*B*) *microplus* by standard PCR and sequencing using the *23S* Anaplasmataceae gene amplifying a 520-pb fragment of rRNA (Table 3). A BLAST analysis showed that four of the sequences obtained were 100% identical to the corresponding sequence of *Anaplasma marginale* (Genbank: CP023731), one of sequences obtained was 100% identical to the corresponding sequence of *Ehrlichia rustica* (Genbank: KT364330), one was 99.13% identical to the corresponding sequence of *Anaplasma phagocytophilum* (Genbank: CP015376) and one was 100% identical to the corresponding sequence of *Anaplasma platys* (Genbank: CP046391).

DNA of Piroplasmida was detected in 19/361 (5.3%) of ticks by qPCR using the *5.8S* rRNA gene. Of these, ten (53%) were found in *Rh. sanguineus* s.l. and nine (47%) were found in *Rh.*

**Table 3. Microorganisms detected using molecular biology tools in ticks collected in Vietnam.**

| Microorganisms tested | Tick species | | | Total |
|---|---|---|---|---|
| | *Rh. sanguineus* | *Rh. (Bo) microplus* | *Amblyomma* sp. | |
| *Anaplasmataceae* | **2% (5/251)** | **13.1% (13/99)** | - | **5% (18/361)** |
| *Anaplasma phagocytophilum* | - | 1% (1/99) | - | 0.3% (1/361) |
| *Anaplasma platys* | 0.4% (1/251) | - | - | 0.3% (1/361) |
| *Anaplasma marginale* | 1.2% (3/251) | 1% (1/99) | - | 1.1% (4/361) |
| *Ehrlichia rustica* | - | 1% (1/99) | - | 0.3% (1/361) |
| *Piroplasmida* | **4% (10/251)** | **10.1% (9/99)** | - | **5.3% (19/361)** |
| *Babesia vogeli* | 3.6% (9/251) | - | - | 2.5% (9/361) |
| *Theileria sinensis* | - | 6.1% (6/99) | - | 1.7% 96/361) |
| *Theileria orientalis* | - | 3% (3/99) | - | 0.8% (3/361) |
| *Rickettsia* sp. | - | - | 100% (1/1) | 0.3% (1/361) |
| *Borrelia* sp. | 0.4% (1/251) | - | - | 0.3% (1/361) |

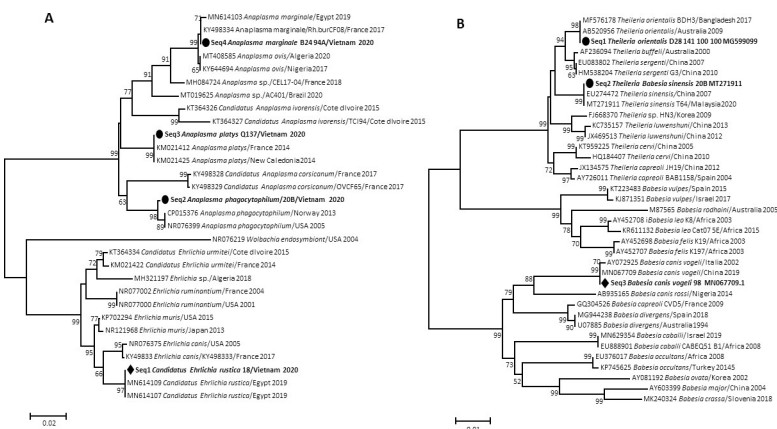

**Fig 4. *23S* rRNA gene-based phylogenetic analysis of strains identified in this study.** Phylogenetic tree highlighting the position of *A. phagocytophilum*, *A. marginale*, *A. platys*, and *E. rustica* identified in our study are close to their homologues available in GenBank (A). *18S* rRNA gene-based phylogenetic analysis of strains identified in the present study. Phylogenetic tree highlighting the position of *B. vogeli*, *T. sinensis*, and *T. orientalis* relative to their correspondence available in GenBank (B).

(*B*) *microplus*. We successfully obtained 18 (95%) sequences by standard PCR and sequencing using the *18S* rRNA gene amplifying a 969-pb fragment of rRNA. The BLAST analysis of nine sequences obtained from *Rh. sanguineus* s.l. revealed that they were between 99.75% and 100% identical to the corresponding sequence of *Babesia vogeli* (GenBank: MN067709), six sequences obtained from *Rh.* (*B*) *microplus* were between 99.82% and 100% identical to the corresponding sequences of *Theileria sinensis* (GenBank: KF559355, MT271911, AB000270) and three sequences obtained from *Rh.* (*B*) *microplus* were between 99.88 and 100% identical to the corresponding sequences of *Theileria orientalis* (GenBank: MG599099) (Table 3).

*Rickettsia* and *Borrelia* sp. were detected by qPCR in one tick of *Amblyomma* sp. and one of *Rh. sanguineus* s.l., respectively. However, all the standard PCR procedures for the identification of *Rickettsia* and *Borrelia* species failed. Of the 25 ticks for which we obtained sequences of microorganisms, 16 (64%) came from engorged ticks and one tick (4%) was co-infected with *A. phagocytophilum* and *T. sinensis*. The species of microorganism, the species of tick and the state of engorgement of the ticks in which the microorganisms were detected are listed in S1 Table.

Two phylogenetic trees of Anaplasmataceae and Piroplasmida were built from the *23S* rRNA and *18S* rRNA genes sequences of our amplicons, respectively. These phylogenetic trees showed that the microorganisms detected in this study are close to their homologues available in GenBank (Fig 4A and 4B).

## Discussion

The correct identification of tick species and associated pathogens can contribute to improving vector control efforts adapted to the surveillance and prevention of outbreaks of tick-borne diseases. In this study, our ticks were identified using traditional methods (morphological) and then confirmed by molecular methods and MALDI-TOF MS, and the associated pathogens were researched using molecular tools. In this study, we combined these three tools to identify ticks and to search for microorganisms associated with these ticks collected in Vietnam.

In this study, the morphological identification of ticks collected in Vietnam revealed six species, including *Rh. sanguineus* s.l., *Rh.* (*B*) *microplus*, *Am. varanensis*, *Amblyomma* sp., *D.*

*auratus* and *D. compactus*. All these species had already been reported in Vietnam [3, 19, 25] and neighbouring countries including Laos, Malaysia, Cambodia, and Thailand [3, 23, 43]. Among the *Rh. sanguineus* s.l. were the species most commonly found on dogs in Vietnam. This tick species is the most widely distributed worldwide and is known to be a vector of several pathogens such as *Anaplasma*, *Rickettsia*, *Ehrlichia*, and *Babesia* spp. [44, 45]. *Rhipicephalus* (*Boophilus*) *microplus* was collected from both cows and goats and is responsible for the transmission of livestock pathogens [6, 24]. There have been several reports of tick-borne livestock pathogens such as *Anaplasma* spp., *Ehrlichia ruminantium*, *Babesia bigemina*, *Babesia bovis*, and *Theileria* spp. [46–48]. However, this tick rarely bites humans [22]. Other tick species were collected from wild animals (pangolins and pigs). Several species of ticks of the genus *Amblyomma* have been collected from almost all species of pangolins [49, 50] and are vectors of *Rickettsia*, *Ehrlichia* spp. [51]. Recently, several studies reported *Amblyomma javanense* detected from pangolins in Singapore [52] and China [53], and *Amblyomma compressum* ticks on pangolins from Congo [54]. Our study is the first to observe *Am. varanensis*, *Amblyomma* sp. on pangolins from Vietnam. *Dermacentor auratus*, *D. compactus* are widely distributed across Sri Lanka, Bangladesh, India, and SEA including Vietnam [55, 56], and are well known vectors of *Rickettsia*, *Coxiella burnetii*, *Borreli*a, and *Anaplasma* spp. [57, 58].

Molecular techniques were used to confirm our morphological identification of tick species by amplifying a portion sequence of a 465-bp fragment *16S* rRNA gene. The choice of the *16S* rRNA gene was based on previous studies that reported that this gene was a reliable tool for tick identification [29, 59]. Interrogating the GenBank database with *16S* rDNA sequences from *Rh. sanguineus* s.l and *Rh.* (*B*) *microplus* showed similarity with the reference sequences available in Genbank for these species that were stored in 70% alcohol for approximately two years. Conversely, we were unable to obtain sequences for all specimens that had been preserved for more than 10 years in alcohol (i.e., *Am. varanensis*, *Amblyomma* sp., *D. auratus*, and *D. compactus*) with the *16S* and *12S* rDNA genes. This might be due to the fact that the alcohol was not completely eliminated during extraction [60] and/or to the fact that these ticks contained blood from their host, which includes several factors that can inhibit the PCR reaction, as already reported [61].

In this study, MALDI-TOF MS was used to identify ticks collected in Vietnam from domestic and wild animals. Among the spectra of tick legs that were subjected to MS analysis, the correct identification rates (LSVs >1.8) were 97%, almost identical to the identification rate reported in other studies [32, 33, 62]. Interestingly, specimens that were not able to be identified by molecular biology were identified by MALDI-TOF MS. This confirms that the tool is reliable and accurate for the identification of ticks. Despite these numerous advantages, this technique is limited by the high cost of the device, although it can be used for clinical microbiology and mycology in addition to entomology, with no additional cost. Maintenance may be another limitation but this can be compensated for by the low cost of reagents once the device is acquired [30]. Secondly, the development of protocols, the choice of the arthropod compartment to be used, the spectra for the creation of the database and, finally, the methods and time of conservation of the arthropods can influence the performance of MALDI-TOF MS [30, 37, 63].

In this study, 10.8% of the ticks were positive for at least one of the microorganisms by qPCR, of which 16/25 (64%) of the ticks carrying DNA of microorganisms by sequencing were engorged ticks. The detection of microorganisms in engorged ticks doesn't have the same epidemiological meaning as when detected in a questing or non-engorged attached tick. Such ticks may potentially have fed on hosts with bacteraemia, thus biasing the estimate of the actual rate of tick infestation.

The microorganisms detected in this study and confirmed by sequencing belong to the Anaplasmataceae family (*A. phagocytophilum*, *A. marginale*, *A. platys*, and *E. rustica*), which

are known aetiologies of zoonotic diseases [8, 13, 64, 65]. The Piroplasmida family (*B. vogeli*, *T. sinensis*, and *T. orientalis*) was mainly known as the potential zoonotic pathogens [66].

*Anaplasma marginale* is responsible for bovine anaplasmosis and is an intracellular bacterium transmitted by tick species mainly belonging to the *Rhipicephalus* and *Dermacentor* genera [67]. The DNA and specific antibodies against *A. marginale* were previously reported in the blood of cattle and cows from Vietnam [23, 24]. This study is the first report of *A. marginale* in *Rh. (B) microplus* and *Rh. sanguineus* s.l ticks collected in Vietnam. However, *A. marginale* had previously been reported in cattle and cattle *Rh. (B) microplus* ticks in China [68], the Philippines [69] which is a neighbouring country to Vietnam, in cattle and cattle ticks in Malaysia [70], and many African countries [71].

*Anaplasma platys*, the aetiological agent of infectious canine cyclic thrombocytopenia and which can be transmitted by *Rh. sanguineus* s.l., *A. platys* has been recorded in China [48], Colombia [72], and detected on various ectoparasites such as *Rh. (B) microplus* [48] and *Hyalomma dromedarii* [73]. *Anaplasma platys* is one of the most significant tick-borne zoonotic pathogens [24, 74] and several cases of human infections have been described in Venezuela [75], Chicago [76], and South Africa [77]. *Anaplasma platys* has already been detected from blood specimens of cattle and dogs in Vietnam [24], but it was the first discovery in *Rh. sanguineus* s.l. ticks from Vietnam in our study. It had been previously detected in *Rh. sanguineus* s.l. in SEA [25], including in the Philippines [78], Thailand, and Malaysia [79, 80].

The pathogen *A. phagocytophilum* is the causative agent of human granulocytic anaplasmosis (HGA) and tick-borne fever in ruminants [81]. It is rarely found in *Rh. (B) microplus* and is known to be transmitted by the *Ixodes* tick genus [82]. Of the detected tick-borne diseases, *A. phagocytophilum* is the most important bacterium due to its wide distribution across Europe, Asia, and North America [83, 84], with several reports of human infections [85, 86]. This is the first study reporting the detection of *A. phagocytophilum* in *Rh. (B) microplus* ticks using the molecular method in Vietnam. It has also been described in the same tick species in China [87] and Malaysia [70].

We found *Candidatus Ehrlichia rustica* in the *Ehrlichia chaffeensis* group, the agent of human monocytic ehrlichiosis [88]. Canine ehrlichiosis was first recorded in a serological study in US military dogs serving in the Vietnam war [89]. The vectors of this pathogen are *Rhipicephalus*, *Amblyomma*, *Dermacentor* spp. [90]. Another study from 2003 reported that *Ehrlichia* spp., which gathered with *E. chaffeensis*, was also discovered in other species, such as *Haemaphysalis hystricis* from wild pigs in Vietnam [22], and *Ixodes sinensis* in China [91].

*Babesia vogeli*, the agent of canine babesiosis in North and South America, is transmitted by *Rh. sanguineus* s.l. and is the less pathogenic species. It is a protozoan found mainly in tropical or subtropical areas of northern, eastern and southern Africa, Asia, and northern and central Australia [92]. In SEA, *B. vogeli* has been described in Malaysia [93] and in the Philippines [94]. The molecular evidence of *B. vogeli* in *Rh. sanguineus* s.l. collected from dogs has been reported in Vietnam [4] and in ticks collected from East and Southeast Asia [25]. The DNA of *B. vogeli* was detected in this study in *Rh. sanguineus* s.l. ticks, confirming the presence of the protozoan in Vietnam.

*Theileria sinensis*, the causative agent of bovine theileriosis, causes economic losses and threats to the cattle industry. *Theileria sinensis* is primarily distributed throughout Asia (including China, the Korean Peninsula, Japan, and Malaysia [95–97]. It was identified in *Haemaphysalis qinghaiensis* ticks collected from cattle and yaks in China [98]. *Theileria* spp. were then detected in *Haemaphysalis longicornis*, *Hyalomma* (i.e., *Hy. detritum*, *Hy. dromedarii*, *Hy. a. anatolicum*, *Hy.a asiaticum*, *Hy. rufipes*), and *Rhipicephalus* sp. [99, 100]. Besides ticks, *Theileria* spp. were also detected in sheep, goat, and ruminant blood samples [101]. This is the first report of *T. sinensis* DNA in *Rh. (B) microplus* in Vietnam.

Similarly, *Theileria orientalis*, the causative agent of oriental theileriosis, is an economically significant protozoan which infects cattle [95]. *Theileria orientalis* is widely distributed in countries such as Japan [102], China [103], Indonesia [104], Australia [105], and New Zealand [95]. The *Theileria orientalis* species has been identified in Vietnam from blood samples from cattle, water buffalo, sheep, goats and *Rh. (B) microplus* ticks collected from these hosts [46]. Here, we showed the presence of 3% *T. orientalis* in *Rh. (B) microplus* collected from cows. Although *Rh. (B) microplus* is not recorded as a vector of *T. orientalis*, none of the common vectors *Amblyomma*, *Dermacentor*, and *Haemaphysalis* spp. [106] were detected in our work.

*Rickettsia* spp. and *Borrelia* spp. detected by qPCR in this study were not amplified and sequenced to confirm their species. As previously reported, this could be caused by the higher sensitivity of qPCR than standard PCR [107].

Co-infections in ticks usually occur after a blood meal from a host co-infected with different microorganisms. In this study, we reported for the first time the co-infection by *A. phagocytophilum* and *T. sinensis* in *Rh. (B) microplus* ticks. The coinfection rate of 0.3% (1/361) in this study is lower those that have been reported in the Côte d'Ivoire [71], and in Mali [33].

## Conclusion

Our work indicates that MALDI-TOF MS is a useful and reliable tool for the identification of alcohol-preserved tick species which have undergone different storage periods collected in Vietnam. Our database demonstrates, for the first time, the prevalence of *A. platys*, *A. phagocytophilum*, *A. marginale*, *E. rustica*, and *T. sinensis* pathogens in ticks collected in Vietnam. Our finding should prompt further investigation to evaluate the potential risks of ticks and tick-associated pathogens in Vietnam. Furthermore, it shows that MALDI-TOF MS may be used as an alternative tool for identifying ticks infected or uninfected by pathogens in future studies.

## Supporting information

**S1 Fig. Flow diagram of tick specimens which were included and analysed using MALDI-TOF MS and molecular tools.**
(TIF)

**S1 Table. The number of microorganisms were detecetd in engorged/non-engored ticks.**
*: Tick was co-infections by two microoganisms.
(DOCX)

## Acknowledgments

We are sincerely grateful to the staff of the Entomological Department of the Institute of Malariology, Parasitology and Entomology, Quy Nhon (IMPE-QN), Vietnam, especially to the staff in the experimental entomology group, for their support with specimen collection and transportation. We would also thank the insectarium team of the Institut Hospitalo-Universitaire Méditerranée Infection, Marseille, France, for their assistance and the provision of the MALDI-TOF MS and molecular biology materials.

## Author Contributions

**Conceptualization:** Ly Na Huynh, Nhiem Le-Viet, Philippe Parola.

**Data curation:** Ly Na Huynh, Adama Zan Diarra, Philippe Parola.

**Formal analysis:** Ly Na Huynh, Adama Zan Diarra, Philippe Parola.

**Funding acquisition:** Philippe Parola.

**Investigation:** Ly Na Huynh, Quang Luan Pham, Nhiem Le-Viet, Van Hoang Ho.

**Methodology:** Ly Na Huynh, Adama Zan Diarra, Xuan Quang Nguyen, Philippe Parola.

**Project administration:** Xuan Quang Nguyen, Philippe Parola.

**Resources:** Ly Na Huynh, Adama Zan Diarra, Jean-Michel Berenger, Philippe Parola.

**Software:** Ly Na Huynh, Adama Zan Diarra.

**Supervision:** Adama Zan Diarra, Philippe Parola.

**Validation:** Ly Na Huynh, Adama Zan Diarra, Jean-Michel Berenger, Philippe Parola.

**Visualization:** Ly Na Huynh, Adama Zan Diarra, Philippe Parola.

**Writing – original draft:** Ly Na Huynh, Philippe Parola.

**Writing – review & editing:** Ly Na Huynh, Adama Zan Diarra, Quang Luan Pham, Nhiem Le-Viet, Jean-Michel Berenger, Van Hoang Ho, Xuan Quang Nguyen, Philippe Parola.

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
