## [Decision Letter · Decision Letter 0]

27 Jun 2021

Dear Pr. Parola,

Thank you very much for submitting your manuscript "Morphological, molecular and MALDI-TOF identification of ticks and tick-associated pathogens in Vietnam" for consideration at PLOS Neglected Tropical Diseases. As with all papers reviewed by the journal, your manuscript was reviewed by members of the editorial board and by several independent reviewers. In light of the reviews (below this email), we would like to invite the resubmission of a significantly-revised version that takes into account the reviewers' comments. 

We cannot make any decision about publication until we have seen the revised manuscript and your response to the reviewers' comments. Your revised manuscript is also likely to be sent to reviewers for further evaluation.

Sincerely,

Lynn Soong, MD, PhD

Deputy Editor

Reviewer's Responses to Questions

**Key Review Criteria Required for Acceptance?**

**Methods**

-Are the objectives of the study clearly articulated with a clear testable hypothesis stated?

-Is the study design appropriate to address the stated objectives?

-Is the population clearly described and appropriate for the hypothesis being tested?

-Is the sample size sufficient to ensure adequate power to address the hypothesis being tested?

-Were correct statistical analysis used to support conclusions?

-Are there concerns about ethical or regulatory requirements being met?

Reviewer #1: Yes

Reviewer #2: - Ethical considerations: you mentioned obtaining authorizations for cows and dogs, but you also sampled goats. What about them?

- Tick collection: How were the wild animal trapped? And then how were they handled? Were they anesthetized? By whom? 

- L170: How confident were you with your morphological identification to create your database solely based on it? Especially since morphological ID of ticks from Vietnam is so challenging. Please make it clear that morphological ID was only used alone when molecular ID wasn’t possible.

- L196: The commonly accepted positivity threshold for bacteria is 35 Ct, not 36. It can be higher for parasites, but you can’t increase it for all your microorganisms. Please correct the sentence and the number of actually positive samples. Also, use the appropriate reference here instead of a self-citation.

**Results**

-Does the analysis presented match the analysis plan?

-Are the results clearly and completely presented?

-Are the figures (Tables, Images) of sufficient quality for clarity?

Reviewer #1: Yes

Reviewer #2: The results are overall well presented and match the analysis plan. I have however a few concerns listed below.

Main comment: it is critical to know which ticks were engorged. Detecting pathogens in a fully engorged tick clearly doesn’t have the same epidemiological meaning as detecting it in a questing or non-engorged attached tick. Please detail how many engorged ticks were included in the study and if the pathogens were detected in engorged/non-engorged ticks. Also, more effort should be put into the molecular identification of ticks.

- L235: Did you check if there was any DNA at all using a nanodrop for example? Ten years isn’t such a long time, DNA has been extracted from much older samples. Maybe some protocol optimization is needed. Have you tried extracting from a smaller sample? You might get some inhibition from the blood; the legs might be a better option. One leg is more than enough.

Table 1: Table 1 is not clear to me, especially the total line. 

- Why is the total of the first column 329? Did you sequence 329 specimens?

**Conclusions**

-Are the conclusions supported by the data presented?

-Are the limitations of analysis clearly described?

-Do the authors discuss how these data can be helpful to advance our understanding of the topic under study?

-Is public health relevance addressed?

Reviewer #1: Yes

Reviewer #2: The limitations of the study are barely addressed. This manuscript describes vector-borne bacteria associated with ticks but don't mention if these ticks were engorged with potentially bacteraemic blood. Moreover, the authors highlighted the superiority of MALDI-TOF MS compared to molecular biology in this particular case but failed to use an optimised molecular biology protocol. The data they are reporting might be of importance but some aspects of this manuscript lack thoroughness.

**Editorial and Data Presentation Modifications?**

Reviewer #1: Require minor review of typographic errors throughout the manuscript.

Reviewer #2: The short title isn’t really shorter that the main title

ABSTRACT AND SUMMARY

- L32: “329 (91%) specimens were of excellent quality” I guess you mean the spectra of these 239 specimens were of excellent quality

- L34: Please indicate median value 

INTRODUCTION

- L66: Ticks are not ectoparasites of pathogens, please correct this sentence

- L67: Arthropods transmit pathogens, not diseases, please correct

- L74: Rephrase. “Despite the perceived economic benefits of livestock farming, the country potentially faces challenges in terms of food safety risks and transmission of zoonotic diseases.”

- L85: Some important references are missing. Kolonin is only cited for the first time in your discussion. He wrote the only review on ticks of Vietnam and should be cited in your introduction. Nguyen et al. reported tick pathogens associated with Rh. sanguineus in Vietnam in 2019. Apanaskevich can also be mentioned for his recent new Vietnamese tick species descriptions.

- L87: “focused” please correct

METHODS

- L187: Please replace “detection of pathogens” by “detection of microorganisms”. You targeted Anaplasmataceae, those primers also amplify Wolbachia sp. which are not pathogens.

RESULTS

- L221: it’s only one tick so I imagine it’s one pangolin, not “pangolins” please correct

- L223: the morphological characteristics of the ticks are barely presented on Fig1. This is just a picture of the different ticks. To show the morphological criteria, please present them like Boucheikhchoukh et al. (CIMID 2018) or just rephrase.

- L242: “91% (329) of 242 specimens had excellent quality” One word is missing here… spectra I imagine?

- L264: “microorganisms” please correct

- L268: “or Bartonella” please correct

- L270: “The DNA of this bacterium” You’re talking about Anaplasmataceae, it’s a family of bacteria

DISCUSSION

- L310: Avoid abbreviating names at the beginning of a sentence. Correct throughout the manuscript

- L311: Again, vectors transmit pathogens, not diseases, please correct

- L315: “The Amblyomma genus is the most common tick species” So are you talking about or a genus or a species?

- L318: I imagine there’s a typo here and you’re talking about A. compressum? 

- L331-332: You can use the remaining legs for molecular biology to limit PCR inhibition

- L337: “to be identified” please correct

- L348: “which are known etiologies of zoonotic diseases” please correct

- L353: “antibodies against A. marginale” please correct

- L354: “the first report” please correct

- L414: “MALDI-TOF MS” please check throughout the manuscript that you’ve added “MS” after “MALDI-TOF”

- L408: Please cite an original paper reporting the difference of sensitivity between qPCR and standard PCR. Diarra et al. simply cited this too.

Author contributions: contribution of Jean Michel Berenger is missing

Reference list: please check format of references 33, 39, 43, 48

TABLES

Table 1:

- Please add to your manuscript that no blind test was performed for D. compactus because of the low number of specimens.

- Dermacentor was abbreviated “D” throughout the manuscript (which is the validated abbreviation), please keep this abbreviation in the table

- Head of column 5: “MALDI-TOF MS”

Table 2: 

- Anaplasmataceae and Piroplasmida shouldn’t be italicized 

- Please add “spp.” after genus names

- Where is the reference for the Borrelia spp. ITS4 primers?

- Please correct “Correlia burnettii”

Table 3: “sp” shouldn’t be italicized 

Dendrogram: Use validated abbreviations for species names

**Summary and General Comments**

Reviewer #1: The manuscript describes a comparative evaluation of morphological, molecular and MALDI-TOF identification of tick species collected in Vietnam and preserved long term in 70% alcohol. Also, identify microorganisms associated with those ticks using molecular methods. Results indicated that the MALDI-TOF is a potential useful and reliable tool for the identification of alcohol-preserved tick species.

Reviewer #2: This manuscript by Philippe Parola and co-authors is another confirmation of the reliability of MALDI-TOF MS for the identification of ticks collected in the field. It also reports pathogenic microorganisms in these ticks in Vietnam for the first time. Considering the scarcity of studies on tick-borne diseases in Vietnam and neighbouring countries, such data is always welcome. Nevertheless, a few things need to be addressed. The authors really need to clarify which pathogens were detected from engorged ticks as the epidemiological hypotheses are very different from those we formulate when studying questing ticks. I would also like to see some effort from the authors to refine their molecular biology assays. We all struggle sometimes to sequence some specimens, but extracting the DNA from tick halves (possibly engorged ones)and stopping there doesn't look like much effort. I added some suggestions in my specific comments but many others are available in the literature. On another note, I also think the authors should pay attention to the number of self-citations. I am not referring to MALDI-TOF MS papers, as I am well aware that the authors were among the first ones to apply this technique to entomology. The authors tend however to not cite appropriate references but instead cite their previous papers where they made similar statements. This is not appropriate and needs to be corrected throughout the manuscript. Finally, the English of the manuscript really needs to be improved.

PLOS authors have the option to publish the peer review history of their article (what does this mean?). If published, this will include your full peer review and any attached files.

Reviewer #1: No

Reviewer #2: No
---

## [Editor Report · Decision Letter 1]

13 Sep 2021

Dear Pr. Parola,

We are pleased to inform you that your manuscript 'Morphological, molecular and MALDI-TOF MS dentification of ticks and tick-associated pathogens in Vietnam' has been provisionally accepted for publication in PLOS Neglected Tropical Diseases.

Best regards,

Lynn Soong, MD, PhD

Deputy Editor

---

## [Editor Report · Acceptance letter]

23 Sep 2021

Dear Pr. Parola,

We are delighted to inform you that your manuscript, " Morphological, molecular and MALDI-TOF MS identification of ticks and tick-associated pathogens in Vietnam," has been formally accepted for publication in PLOS Neglected Tropical Diseases.

Best regards,

Shaden Kamhawi

co-Editor-in-Chief

Paul Brindley

co-Editor-in-Chief
